# Mitochondrial dysfunction induces RNA interference in *C. elegans* through a pathway homologous to the mammalian RIG-I antiviral response

Kai Mao[1,2], Peter Breen[1,2], Gary Ruvkun [1,2]*

1 Department of Molecular Biology, Massachusetts General Hospital, Boston, Massachusetts, United States of America, 2 Department of Genetics, Harvard Medical School, Boston, Massachusetts, United States of America

* ruvkun@molbio.mgh.harvard.edu

**Data Availability Statement:** All relevant data are within the paper and its Supporting Information files.

## Abstract

RNA interference (RNAi) is an antiviral pathway common to many eukaryotes that detects and cleaves foreign nucleic acids. In mammals, mitochondrially localized proteins such as mitochondrial antiviral signaling (MAVS), retinoic acid-inducible gene I (RIG-I), and melanoma differentiation-associated protein 5 (MDA5) mediate antiviral responses. Here, we report that mitochondrial dysfunction in *Caenorhabditis elegans* activates RNAi-directed silencing via induction of a pathway homologous to the mammalian RIG-I helicase viral response pathway. The induction of RNAi also requires the conserved RNA decapping enzyme EOL-1/DXO. The transcriptional induction of *eol-1* requires DRH-1 as well as the mitochondrial unfolded protein response (UPR$^{mt}$). Upon mitochondrial dysfunction, EOL-1 is concentrated into foci that depend on the transcription of mitochondrial RNAs that may form double-stranded RNA (dsRNA), as has been observed in mammalian antiviral responses. Enhanced RNAi triggered by mitochondrial dysfunction is necessary for the increase in longevity that is induced by mitochondrial dysfunction.

## Introduction

Many RNA viruses carry an RNA-dependent RNA polymerase (RdRP) to replicate their RNA genome in the host, bypassing entirely information storage and replication with DNA. In this way, RNA viruses can replicate in nondividing cells with lower deoxyribonucleotide levels than those required for DNA viruses or retroviruses but with the substantial ribonucleotide levels needed for transcription of RNA and mRNAs in terminally differentiated cells. A double-stranded RNA (dsRNA) viral replication intermediate is a strong clue to the host cell that an RNA virus infection is underway. Recognition of the dsRNA replication intermediate is an initial step in antiviral immune responses. In mammals, the RNA helicases retinoic acid-inducible gene I (RIG-I) or melanoma differentiation-associated protein 5 (MDA5) recognize the RNA signatures of RNA virus replication [1]. RIG-I binds to dsRNA, as well as single-

**Funding:** K.M. is a Damon Runyon Fellow supported by the Damon Runyon Cancer Research Foundation (DRG-2213-15). This work is supported by a grant from the National Institute of Health awarded to G.R. (NIH GM044619 and AG16636). The funders had no role in study design, data collection and analysis, decision to publish, or preparation of the manuscript.

**Competing interests:** The authors have declared that no competing interests exist.

**Abbreviations:** CARD, caspase activation and recruitment domains; CTD, C-terminal domain; dsRNA, double-stranded RNA; Eri, enhanced RNAi; ETC, electron transport chain; gRNA, guide-RNA; lir, *lin-26*-related; MAVS, mitochondrial antiviral signaling; MDA5, melanoma differentiation-associated protein 5; m$^7$G, N$^7$-methyl guanosine; NAD, nicotinamide adenine dinucleotide; NTD, N-terminal domain; PPH, pyrophosphohydrolase; PPi, pyrophosphate; RdRP, RNA-dependent RNA polymerase; RIG-I, retinoic acid-inducible gene I; RNAi, RNA interference; rRNA, ribosomal RNAs; RT-qPCR, quantitative reverse transcription PCR; siRNA, small interfering RNA; ssRNA, single-stranded RNA; tRNA, transfer RNAs; UPR$^{mt}$, unfolded protein response.

stranded RNA (ssRNA) with 5′-triphosphate that is a signature of the products from RdRPs that mediate viral RNA replication [2]. RIG-I or MDA5 associate with the mitochondrial antiviral signaling protein (MAVS) and elicit the downstream nuclear factor kappa B (NF-κB) and other interferon immune signaling that mediate systemic antiviral immune defenses [3].

The nematode *Caenorhabditis elegans* uses an RNA interference (RNAi) pathway to mediate antiviral defense instead of the interferon signaling of vertebrates and most invertebrates [4–7]. RNAi is a highly conserved mechanism for antiviral defense and broadly deployed by eukaryotes, including, fungi, nematodes, insects, plants, and vertebrates [8–10]. RNAi was initially discovered to mediate silencing triggered by engineered dsRNA triggers but soon discovered to produce natural small interfering RNAs (siRNAs) that target specific mRNAs for degradation and chromatin regions for epigenetic silencing. RNAi is mediated by 22 nt to 26 nt single-stranded short interfering or siRNAs that are produced and presented to target mRNAs by the Argonaute proteins and the Dicer dsRNA ribonuclease, conserved across many but not all eukaryotes [11].

Some of the genes in the vertebrate and insect NF-κB pathway are conserved between nematodes and other animals without RdRP genes. Like its mammalian orthologues RIG-I and MDA5, the *C. elegans* DRH-1 mediates antiviral RNAi and is essential for neutralization of an invading virus [12,13]. DRH-1 was initially identified as a binding partner with the Dicer protein DCR-1, the Argonaute protein RDE-1, and the RNA helicase RDE-4 [14]. In addition to *drh-1*, *drh-2* is a pseudogene, and *drh-3* encodes a component of an RdRP protein complex [15].

The DCR-1 ribonuclease is the first step in siRNA generation in most of the *C. elegans* RNAi pathways, including exogenous RNAi, endogenous RNAi, and antiviral RNAi [15,16]. siRNAs generated by these pathways engage the Mutator proteins that mediate siRNA-guided repression in both exogenous and endogenous RNAi [17,18]. Mutations in the Dicer, Argonaute, and accessory factors that generate and present siRNAs to target mRNAs are generally defective for RNAi or antiviral defense. But the Argonaute and RdRP gene families are expanded in *C. elegans* compared with most animals, and surprisingly, loss-of-function mutations in some of those genes causes an increase in the response to siRNAs: mutations in the RdRP RRF-3, the specialized Argonaute ERGO-1, the RNA helicase ERI-6/7, or the exoribonuclease ERI-1 enhance silencing by siRNAs [19,20]. Many of these enhanced RNAi (Eri) mutations desilence retroviral elements in the *C. elegans* genome so that antiviral response pathways are triggered to in turn induce the expression of RNAi-based antiviral defense [21].

A mutation in the *C. elegans* mitochondrial chaperone gene *hsp-6* causes induction of a suite of drug detoxification and defense genes [22]. Here, we show that in addition to the induction of these defense pathways, RNAi pathways are also activated. Because RNAi is a key feature of *C. elegans* antiviral defense, and because of the association of mammalian viral defense pathways such as MAVS and RIG-I with the mitochondrion, we explored more fully how mitochondrial homeostatic pathways connect to RNAi and antiviral defense in *C. elegans*. We find that reduction of function mutations in a wide range of mitochondrial components robustly enhanced RNAi-mediated silencing of endogenous genes as well as a variety of reporters of RNAi. These antiviral responses to mitochondrial dysfunction are homologous to the RIG-I-based mitochondrial response in mammals because they depend on the RIG-I homologue, the DRH-1 RNA helicase. Comparing the *C. elegans* transcriptional response of a mitochondrial mutant and infection with the Orsay RNA virus, we found a striking overlap of expression of multiple members of *C. elegans* *pals* genes implicated in antiviral and anti-pathogen response pathways [23,24] and the *eol-1*/DXO RNA decapping enzyme gene. We found that *eol-1* transcription is dramatically induced by mitochondrial dysfunction, and an *eol-1* null mutation strongly suppresses the DRH-1-mediated antiviral RNAi response normally

induced by mitochondrial dysfunction. We showed that the EOL-1 protein forms foci in the cytosol only if the mitochondrion is stressed, and the production of these foci is dependent on transcription of RNA from the mitochondrial genome. This is reminiscent of the central role that dsRNA released from the mitochondria plays in mammalian antiviral response pathways: During a viral infection, mitochondrial disruption by MAVS and other mitochondrial-associated viral immunity factors releases mitochondrial dsRNA and DNA to the cytosol, where dsRNAs trigger an MDA5-dependent interferon response [25,26], and DNA via cGAS and Sting activate NF-κB immune signaling [27].

Gene inactivation or mutations in a wide variety of nuclear-encoded mitochondrial genes cause one of the strongest increases in the life span in genome screens for life span extension [28]. We find that the uncoupling of the antiviral defense response via *drh-1* or *eol-1* mutations abrogates the increase in life span, suggesting that the antiviral axis of mitochondrial dysfunction is critical to the life span extension.

## Results

### *C. elegans* mitochondrial mutations enhance RNAi using the RNA helicase DRH-1

The *hsp-6(mg585)* allele is a reduction of function mutation (P386S) in the mitochondrial HSP70 chaperone in a region that is conserved between mammals and *C. elegans* (VQEIFGKV**P**SKAVNPDEAVA). *hsp-6(mg585)* causes induction of a suite of drug detoxification and defense genes that are also induced by a variety of mitochondrial mutations or toxins that disrupt mitochondrial function [22]. *C. elegans* HSP-6 and human mtHSP70 are orthologues of bacterial and archaeal dnaK; these mitochondrial chaperones were bacterial chaperones before the mitochondrial endosymbiosis event more than a billion years ago and the migration of these bacterial genes to the eukaryotic nuclear genome [22]. Null alleles of *hsp-6* cause developmental arrest, presumably because of the defects in the folding or import of many mitochondrial client proteins [29,30]. But the viable *hsp-6(mg585)* allele allows mitochondrial roles in other pathways to be studied without the associated lethality. In addition to activating detoxification and immune responses, we found that the mitochondrial defects caused by *hsp-6(mg585)* also unexpectedly cause enhanced RNAi, which is an antiviral defense pathway.

RNAi in *C. elegans* can be induced by ingestion of approximately 1 kb of dsRNA corresponding to a particular gene. For feeding RNAi, *Escherichia coli* have been engineered to produce any of 18,000 dsRNAs corresponding to any *C. elegans* gene [31]. The specificity of RNAi in *C. elegans* is superior to the single siRNA approaches common in most animal systems, probably because, almost unique to animals, *C. elegans* uses RdRP genes to amplify the primary siRNAs produced by the Argonaute and Dicer proteins that nearly all eukaryotes also use for RNAi, and unlike in other animal systems, RNAi in *C. elegans* can be induced by 1-kb segments of dsRNA without the induction of interferon-related responses to dsRNA. Thus, thousands of siRNAs produced from feeding *C. elegans* with an *E. coli* expressing a 1-kb dsRNA sum for on-target effects on target mRNA inactivation and average for off-target mRNA inactivations. The specificity of dsRNA-induced RNAi was validated with genetic loci that had previously studied by genetics: Most dsRNAs corresponding to those genes with known phenotypes recapitulated the phenotypes predicted from genetics [32]. But a minority of genes expected to generate easily scored phenotypes by RNAi did not silence in wild type. These dsRNAs were used in genetic screens for *C. elegans* mutations that enhance RNAi, or *eri-* mutations [19]. For example, mutations in the conserved exonuclease *eri-1* or the RdRP *rrf-3* cause Eri [15,19]. Most mutants that enhance RNAi responses also cause silencing of

multicopy transgenes, because the transgene is detected as a bearing some foreign signatures (lack of introns is a major signature of a viral origin) and silenced in the Eri state of these mutant strains. A set of dsRNA tester genes have been developed that cause strong phenotypes in Eri mutants, but no RNAi phenotype in wild type [33].

We found that the mitochondrial *hsp-6(mg585)* mutant showed the strong phenotype seen in Eri mutants on many of these Eri tester dsRNAs expressed from *E. coli*. For example, a dsRNA targeting the *lir-1* (where *lir* is an abbreviation for *lin-26*-related) gene causes no phenotype in wild type, but causes lethal arrest in, for example, the *eri-6(mg379)*-Eri mutant, as well as on the mitochondrial *hsp-6(mg585)* mutant (Fig 1A). The enhanced lethality after exposure to *lir-1* dsRNA in Eri mutants is due to siRNAs from this dsRNA targeting the duplicated and diverged genes with high regions of nucleotide homology on the same primary transcript of the *lir-1*, *lir-2*, and *lin-26* operon [34]. One explanation for the enhanced response to *lir-1* RNAi is that these genes have genetic interactions with *hsp-6(mg585)* that caused synthetic enhancement of certain phenotypes. Therefore, we tested whether other mitochondrial mutations enhance response to Eri tester dsRNAs. *lir-1* RNAi also caused a lethal arrest on a variety of other nuclearly encoded mitochondrial protein point mutants, including nicotinamide adenine dinucleotide (NADH) dehydrogenase *nuo-6*/NDUFB4, ubiquinone biosynthesis *clk-1*/COQ7, and the iron–sulfur cluster *isp-1*/UQCRFS1 (Fig 1A). A distinct dsRNA that targets a

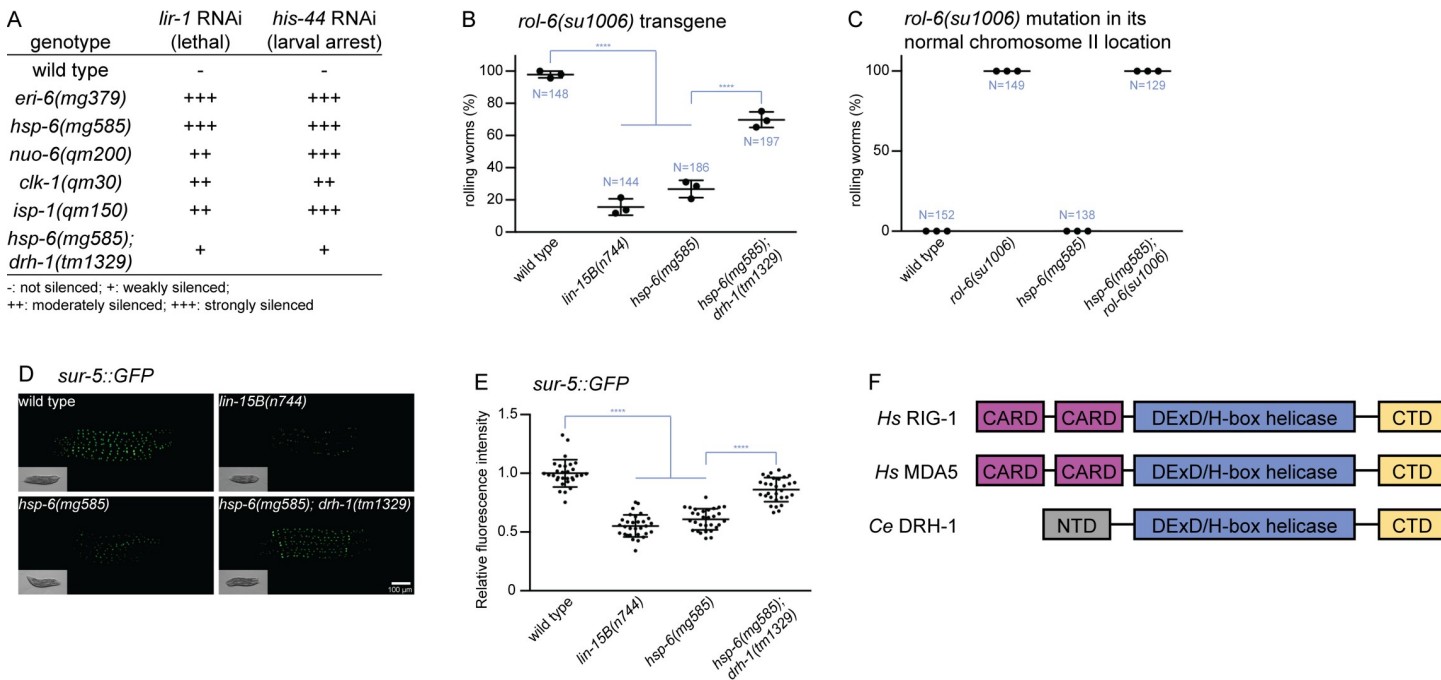

**Fig 1. Mitochondrial mutants activate Eri through DRH-1.** (A) Eri response to *lir-1* RNAi or *his-44* RNAi in the mitochondrial mutants *hsp-6*, *nuo-6*, *clk-1*, *or isp-1* or the *eri-6* Eri mutant causes lethality/arrest but not on wild type. The Eri of the *hsp-6* mitochondrial mutant is suppressed by a *drh-1* mutation. (B) Transgene silencing test with *rol-6(su1006)* transgene. This multicopy transgene in wild type causes a rolling behavior due to the expression of an amino acid substitution in the *rol-6* collagen gene. The expression of the *rol-6* collagen mutation from the multicopy transgene is silenced by Eri in mitochondrial mutants, and this transgene silencing depends on *drh-1* gene activity. Results of 3 replicate experiments are shown. *N*: total number animals tested; **** denotes $p < 0.0001$. (C) The mitochondrial mutations do not simply suppress the collagen defect of a *rol-6* mutation: If the *rol-6(su1006)* mutation is chromosomally expressed, not from a transgene that is under RNAi control, the Rol phenotype is not suppressed by *hsp-6(mg585)* and an *hsp-6(mg585)* single mutation does not have any Rol phenotype. Results of 3 replicate experiments are shown. *N*: total number of animals tested. (D and E) Transgene silencing test with *sur-5*::*GFP* transgene. This transgene is ubiquitously expressed in all somatic cells. The expression of the *sur-5*::*GFP* from the transgene is silenced by Eri in mitochondrial mutants, and this transgene silencing depends on *drh-1* gene activity. Animals were imaged in (D) and the fluorescence was quantified in (E). **** denotes $p < 0.0001$. (F) Diagrams of human RIG-I, MDA5, and *C. elegans* DRH-1. The helicase domain and CTD are conserved. The underlying numerical data can be found in S1 data. CARD, caspase recruitment domain; *Ce*, *Caenorhabditis elegans*; CTD, C-terminal domain; Eri, enhanced RNAi; *Hs*, Homo sapiens; NTD, N-terminal domain; RNAi, RNA interference.

histone 2B gene, *his-44*, causes larval arrest in Eri mutants but no lethality on wild type [35] and also caused larval arrest in the *hsp-6*/mtHSP70, *nuo-6*/NDUFB4, *clk-1*/COQ7, and *isp-1*/ UQCRFS1 mitochondrial mutants (Fig 1A). *his-44* maps to a cluster of histone genes including multiple histone 2B genes with nucleic acid homology; in the Eri mutants, the initial siRNAs produced from the *his-44* dsRNA may spread to adjacent histone 2B genes. Thus, the enhanced response to *lir-1* or *his-44* RNAi was not due to strain background mutations in *hsp-6(mg585)* or the pleiotropy of a mitochondrial chaperone that may affect the function of many imported mitochondrial proteins. Three other mitochondrial point mutants that we tested (out of 6 tested), *nduf-7(et19)*, *mev-1(kn1)*, and *gas-1(fc21)* did not cause an Eri phenotype (S1A Fig). Many mutations in nuclearly encoded mitochondrial proteins are lethal, so that amino acid sub- stitution reduction in function alleles that we tested constitutes most of the available viable mitochondrial mutations. NUO-6, GAS-1, and NDUF-7 are components of complex I, MEV-1 is from complex II, ISP-1 is a component of complex III, and CLK-1 produces the ubiquinone that also functions in electron transport. There was no obvious distinction between the types of mitochondrial mutations that caused Eri and those that did not, in terms of growth rate or severity of the phenotype. It is possible that only particular mitochondrial insults activate the RNAi pathway. The induction of Eri by multiple mitochondrial mutations favored the hypothe- sis that many but not all mitochondrial defects trigger an Eri phenotype.

A hallmark of an Eri phenotype is the silencing of transgenes, as increased RNAi detects the foreign genetic signatures (the fusion of non-*C. elegans* genes such as green fluorescent protein (GFP), the synthetic introns, and other engineered features) of transgenes [36]. We asked if the *hsp-6(mg585)* mutant displayed enhanced transgene silencing by introducing a transgene containing *rol-6(su1006)*, a dominant mutation of a hypodermal collagen that causes an easily scored rolling movement phenotype [37]. *lin-15B* is a class B synthetic multivulva (synMuv B) gene, and loss of function of *lin-15B* significantly enhances transgene silencing [38]. A chro- mosomally integrated transgene carrying multiple copies of the *rol-6(su1006)* mutant collagen gene causes 100% of transgenic animals to roll (a Rol phenotype) in wild-type animals, but in strains with enhanced RNAi, this transgene is now silenced so that 16% of animals are Rol in the *lin-15B(n744)* mutant and 27% are Rol in the *hsp-6(mg585)* homozygous mutant (Fig 1B). The suppression of the Rol phenotype from the transgene carrying *rol-6(su1006)* was not due to a genetic interaction between *hsp-6(mg585)* and the *rol-6(su1006)* mutant collagen gene, because the *hsp-6(mg585)*; *rol-6(su1006)* double mutant with the *rol-6* collagen mutation located in its normal chromosomal location not subject to Eri silencing of a transgene, was still 100% Rol (Fig 1C). Rather, the Eri of the mitochondrial mutants, including *hsp-6(mg585)*, causes a silencing of the *rol-6(su1006)* mutant collagen allele on the multicopy transgene to suppress the Rol phenotype.

To evaluate transgene silencing in other tissues, a ubiquitously expressed *sur-5::GFP* fusion gene was monitored in all somatic cells [39]. The bright GFP signal of *sur-5::GFP* was dramati- cally decreased in the Eri mutant *lin-15B(n744)* as well as in the *hsp-6(mg585)* mitochondrial mutant (Fig 1D and 1E). Thus, *hsp-6(mg585)* enhances exogenous RNAi and silences somatic transgenes. The somatic transgene silencing and Eri in the *eri-1* mutant is associated with a failure to nuclearly localize the Argonaute transcriptional silencing factor NRDE-3 that acts downstream of siRNA generation [33,40]. The synMuvB Eri mutants cause a somatic misex- pression of the normally germline P granules implicated in siRNA [40]. However, neither NRDE-3 nuclear delocalization nor the somatic expression of P granules occurred in *hsp-6 (mg585)* mutant, suggesting that mitochondrial dysfunction does not induce the *eri-1/rrf-3* or synMuvB classes of Eri.

The Eri of *hsp-6(mg585)* most resembled the *eri-6/7* RNA helicase and *ergo-1* Argonaute Eri phenotypes, associated with desilencing of recently acquired viral genes and induction of viral

immunity, without nuclear NRDE-3 or somatic P granule expression [21,36,41]. In mammalian cells, the RNA helicase MDA5 mediates an interferon antiviral immune response that is strongly enhanced by a mitochondrial RNA degradation mutation that enhances production of mitochondrial dsRNAs [25]. Intriguingly, mutations in the *C. elegans* homologue of MDA5, *drh-1*, suppress the synthetic lethality of Eri and dsRNA editing double mutants [21,42]. *C. elegans* DRH-1 contains 3 domains, including conserved helicase domain and C-terminal domain (CTD) (Fig 1F) and an N-terminal domain (NTD) that is only conserved in nematodes (S1A Fig). The caspase activation and recruitment domains (CARD) of RIG-I and MDA5 associate with the mitochondrial protein MAVS, which is unique to mammals [3]. Conversely, the NTD of DRH-1 is nematode-specific and essential for the inhibition of viral replication [13]. To test if DRH-1 is required for *hsp-6(mg585)* induced transgene silencing, 2 transgenes *rol-6(su1006)* and *sur-5::GFP* were tested for silencing in an *hsp-6*; *drh-1* double mutant. The *drh-1(tm1329)* allele disrupts the NTD and renders the strain susceptible to viral infection [12]. *drh-1(tm1329)* suppressed the transgene silencing induced by *hsp-6(mg585)*: *hsp-6(mg585)* animals carrying the *rol-6(su1006)* transgene were 27% Rol, but the *hsp-6 (mg585)*; *drh-1(tm1329)* double mutant was 70% Rol (Fig 1B), showing that the transgene silencing induced by *hsp-6(mg585)* was suppressed by *drh-1(tm1329)*. Similarly, the GFP fluorescence of *sur-5::GFP* was significantly brighter in the *hsp-6(mg585)*, *drh-1(tm1329)* double mutant compared with *hsp-6(mg585)* (Fig 1D and 1E). The *lir-1* and *his-44* Eri lethal or larval arrest phenotypes of *hsp-6(mg585)* were also suppressed by *drh-1(tm1329)* (Fig 1A). *drh-1* mutant animals carrying a distinct mutant allele from a wild strain of *C. elegans* with a C-terminal truncation are competent for RNAi but show defects in antiviral RNAi [16]. Therefore, *hsp-6(mg585)* enhancement of transgene silencing and *his-44* and *lir-1* RNAi responses require DRH-1 gene activity.

## EOL-1 acts downstream of DRH-1 for somatic silencing

Upon virus infection, human RIG-I and MDA5 mediate the up-regulation of interferon genes. Although the interferon signaling pathway is not conserved in *C. elegans*, we suspected DRH-1, like its mammalian orthologues, might promote the transcriptional activation of downstream response genes. Comparison of the expression profile of *hsp-6(mg585)* [22] and wild-type animals infected with the Orsay RNA virus [43] showed that of the 126 genes most up-regulated by Orsay virus infection, 45 genes were also the top up-regulated genes in the *hsp-6 (mg585)* mutant (Fig 2A and S1 Table). Of these 45 genes, 12 are members of the *pals-1* to *pals-40* genes, which located in a few clusters, and remarkably, a null allele in *pals-22* causes an Eri phenotype, multicopy transgene silencing [23,24]. Thus, the increased expression of *pals*-genes in virus-infected and mitochondrial defective animals is not just associated with enhanced RNAi, but a mutation in one *pals-22* gene can cause increased RNAi and transgene silencing.

The *eol-1* RNA decapping gene was also strongly induced in virally infected and mitochondrial mutant *C. elegans* (S1 Table). We selected *eol-1* for detailed analysis because (1) it is conserved from yeast to mammals (Fig 2B); (2) *eol-1* expression is induced by multiple mitochondrial mutations as well as in a *eri-6*; *adr-1/2* Eri mutant that desilences endogenous retroviruses and retrotransposons [21,44]; (3) the induction of *eol-1* by Orsay virus infection is dependent on *drh-1* [45]; and (4) the human orthologue of EOL-1, DXO, represses hepatitis C virus replication [46]. *C. elegans eol-1* was initially identified as a mutant that enhanced olfactory learning after *Pseudomonas aeruginosa* infection [47]. *eol-1* encodes a decapping exoribonuclease orthologous to yeast Rai1 and mammalian DXO (Fig 2B). Mammalian DXO processes the 5′ end of mRNA [48], including a decapping activity that removes the unmethylated

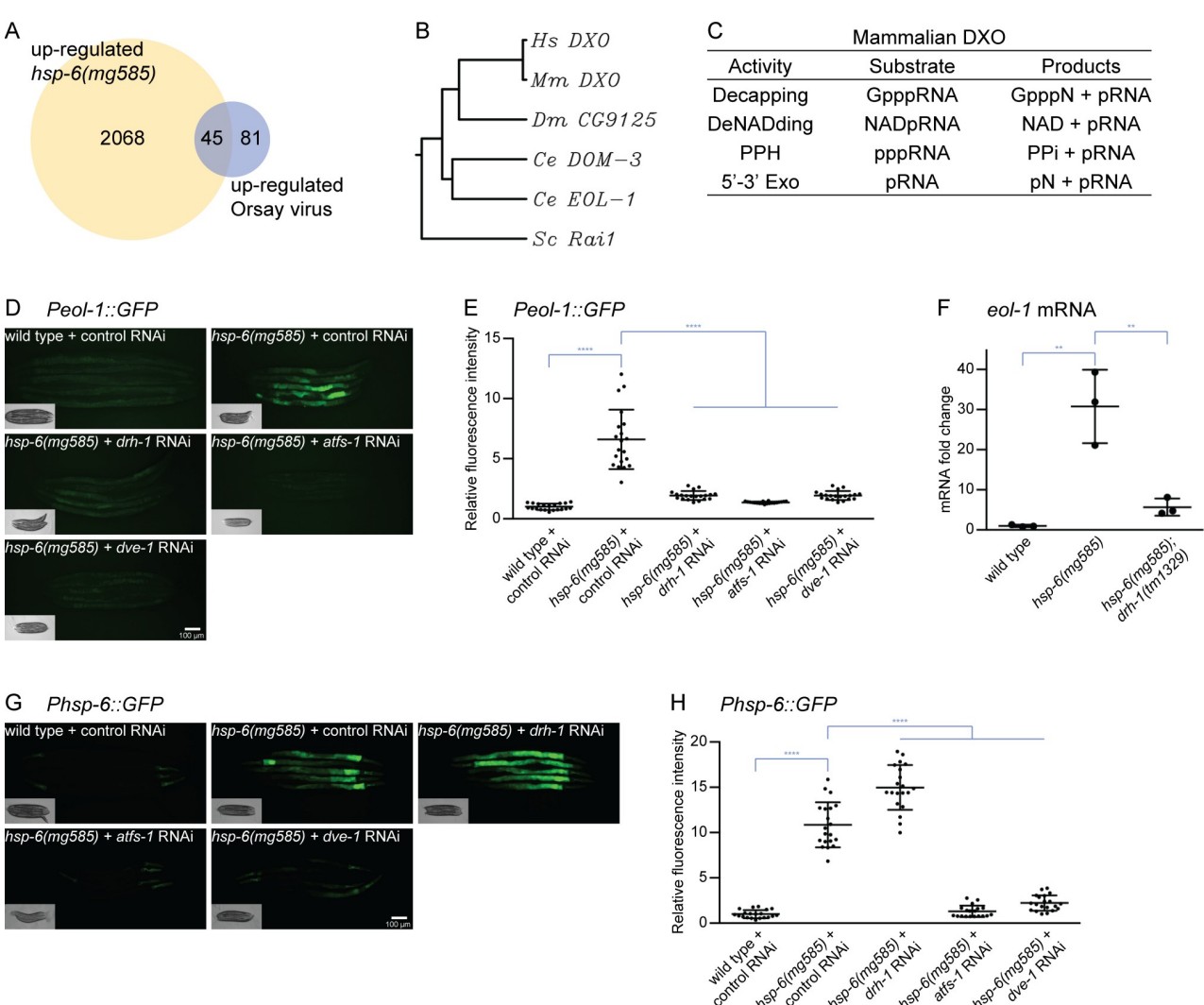

**Fig 2. Increased *eol-1* expression depends on DRH-1.** (A) Venn diagram of genes up-regulated in *hsp-6(mg585)* and Orsay virus infection. (B) Phylogenetic tree of EOL-1/DXO. EOL-1 is conserved from yeast to mammals. (C) The enzyme activities of human DXO. Mammalian DXO modifies the 5′ end of mRNAs: decapping, deNADing, pyrophosphohydrolase and 5′-3′ exonuclease. (D and E) The induction of *Peol-1::GFP* transcriptional fusion reporter requires *drh-1* and UPR^mt. The *Peol-1::GFP* is strongly induced by *hsp-6(mg585)* mitochondrial mutant, and this induction is abrogated by *drh-1* RNAi as well as RNAi of genes involved in UPR^mt (*atfs-1* and *dve-1*). Animals were imaged in (D) and the fluorescence was quantified in (E). **** denotes $p < 0.0001$. (F) The *hsp-6(mg585)* mutant causes increased mRNA level of *eol-1* in a *drh-1*-dependent manner. RT-qPCR assays showed the mRNA level of *eol-1* was induced by *hsp-6(mg585)* mutant, and this induction was abolished in *hsp-6(mg585); drh-1(tm1329)* double mutant. ** denotes $p < 0.01$. (G and H) DRH-1 does not contribute to UPR^mt. The benchmark reporter of UPR^mt *Phsp-6::GFP* is induced by the *hsp-6(mg585)* mutant. And the induction is suppressed by RNAi of *atfs-1* or *dev-1*, but not *drh-1*. Animals were imaged in (G) and the fluorescence was quantified in (H). **** denotes $p < 0.0001$. The underlying numerical data can be found in S1 data. *Ce*: *Caenorhabditis elegans*; *Dm*: *Drosophila melanogaster*; *Hs*: Homo sapiens; *Mm*: *Mus musculus*; RT-qPCR, quantitative reverse transcription PCR; *Sc*: *Saccharomyces cerevisiae*. UPR^mt, mitochondrial unfolded protein response.

guanosine cap and the first nucleotide (GpppN), the deNADing activity that removes the nicotinamide adenine dinucleotide (NAD) cap, and the pyrophosphohydrolase (PPH) activity that releases pyrophosphate (PPi) from 5′ triphosphorylated RNA, and 5′-3′ exonuclease (5′-3′ Exo) to degrade the entire RNA (Fig 2C). The removal of the N^7-methyl guanosine (m^7G) cap and subsequent degradation of mammalian mRNAs are directed by the decapping exoribonuclease DXO. Expression of *Mus musculus* DXO rescued the enhanced olfactory learning phenotype in *C. elegans eol-1* mutant indicating not only the amino acid sequence but also the

function of EOL-1 is conserved [47]. *eol-1* is one of 10 *C. elegans* DXO homologues. Interestingly, 6 of these genes (M01G12.7, M01G12.9, M01G12.14, Y47H10A.3, Y47H10A.4, and Y47H10A.5) are clustered as tandemly duplicated genes, adjacent to *rrf-2*, one of 4 *C. elegans* RdRPs, and one (C37H5.14) is adjacent to *hsp-6*.

To verify the transcriptional up-regulation of *eol-1* in *hsp-6(mg585)* mutant and test if the induction requires DRH-1, a transcriptional fusion reporter *Peol-1::GFP* containing the *eol-1* promoter, GFP, and *eol-1* 3'UTR was constructed. In wild-type animals treated with control RNAi, *Peol-1::GFP* was barely detectable (Fig 2D and 2E); in the *hsp-6(mg585)* mitochondrial mutant, *Peol-1::GFP* was strongly induced (Fig 2D and 2E). The induction of *Peol-1::GFP* was abrogated in *hsp-6(mg585)* treated with *drh-1* RNAi (Fig 2D and 2E). Since the *Peol-1::GFP* transcriptional fusion reporter is a transgene that is subject to Eri transgene silencing by *hsp-6(mg585)* and suppressed by *drh-1*, it might not reflect the accurate expression level of *eol-1* in these mutants. To evaluate the actual mRNA level of chromosomal *eol-1*, quantitative reverse transcription PCR (RT-qPCR) was performed. The mRNA level of chromosomal *eol-1* was 30-fold increased in the *hsp-6(mg585)* single mutant, and this induction was almost completely abolished in the *hsp-6 (mg585); drh-1(tm1329)* double mutant (Fig 2F). Thus, mitochondrial dysfunction of *hsp-6 (mg585)* triggers the transcriptional activation of *eol-1* in a DRH-1-dependent manner.

In *C. elegans*, the mitochondrial unfolded protein response (UPR$^{mt}$) is a transcriptional program responding to mitochondrial dysfunction and essential for mitochondrial recovery, immunity, detoxification, and aging [49]. RNAi of the genes *atfs-1* or *dev-1*, which encode transcription factors that mediate the expression of UPR$^{mt}$ genes [50,51], suppresses the induction of *Peol-1::GFP* (Fig 2D and 2E). We then tested if DRH-1 contributes to the activation of UPR$^{mt}$ by observing the induction of *Phsp-6::GFP*, the canonical reporter of UPR$^{mt}$. The *Phsp-6::GFP* was strongly induced in the *hsp-6(mg585)* mutant and suppressed by RNAi of *atfs-1* or *dev-1* as expected (Fig 2G and 2H). RNAi of *drh-1* in the *hsp-6(mg585); Phsp-6::GFP* strain did not disrupt induction of *Phsp-6::GFP*; in fact, induction of *Phsp-6::GFP* by *hsp-6(mg585)* after *drh-1(RNAi)* was more than in *hsp-6(mg585)* alone, perhaps because *drh-1* inactivation inhibited *hsp-6(mg585)*-induced Eri and transgene silencing. Thus, the transcriptional activation of *eol-1* requires UPR$^{mt}$ signaling, whereas DRH-1 is not part of the general UPR$^{mt}$.

The DRH-1-dependent up-regulation of *eol-1* implies that EOL-1 might act in the enhanced RNAi pathway induction of the *hsp-6(mg585)* mutant. To test this possibility, a loss-of-function mutant *eol-1(mg698)* was generated by CRISPR-Cas9 [52]. In *eol-1(mg698)*, 6 nucleotides "TGATCA," which contains a stop codon and a Bcl I endonuclease recognition sequence to facilitate genotyping, was inserted into the *eol-1* locus and resulted in a "Lys25 to stop" nonsense mutation (Fig 3A). The *rol-6(su1006)* and *sur-5::GFP* transgenes were tested for silencing in the *hsp-6; eol-1* double mutant. For the *rol-6(su1006)* transgene, the *hsp-6 (mg585); eol-1(mg698)* double mutant, showed 82% rolling compared with 27% in the *hsp-6 (mg585)* single mutant (Fig 3B). For the *sur-5::GFP* transgene, the GFP signal in the *hsp-6 (mg585); eol-1(mg698)* double mutant was substantially increased relative to the *hsp-6(mg585)* single mutant (Fig 3C and 3D). Moreover, the Eri phenotype of *hsp-6(mg585)* as assessed using the lethality of the *lir-1* or *his-44* dsRNAs was also suppressed by *eol-1(mg698)* (Fig 3E).

Because mammalian DXO, a homologue of *C. elegans* EOL-1, is involved in mRNA decay, we tested whether EOL-1 mediates the degradation of mRNAs from transgenes. In such a model, *eol-1* might also be required for the somatic transgene silencing caused by other Eri mutants, for example, by the large synMuv B class of Eri mutants. Although *drh-1(tm1329)* slightly suppressed the transgene silencing of a *lin-15B* null mutant, *eol-1* did not contribute to transgene silencing by the synMuvB mutants (Fig 3F–3H). Thus, EOL-1 does not act in the synMuvB RNAi pathway that silences transgenes and is specifically required for mitochondrial dysfunction-induced transgene silencing.

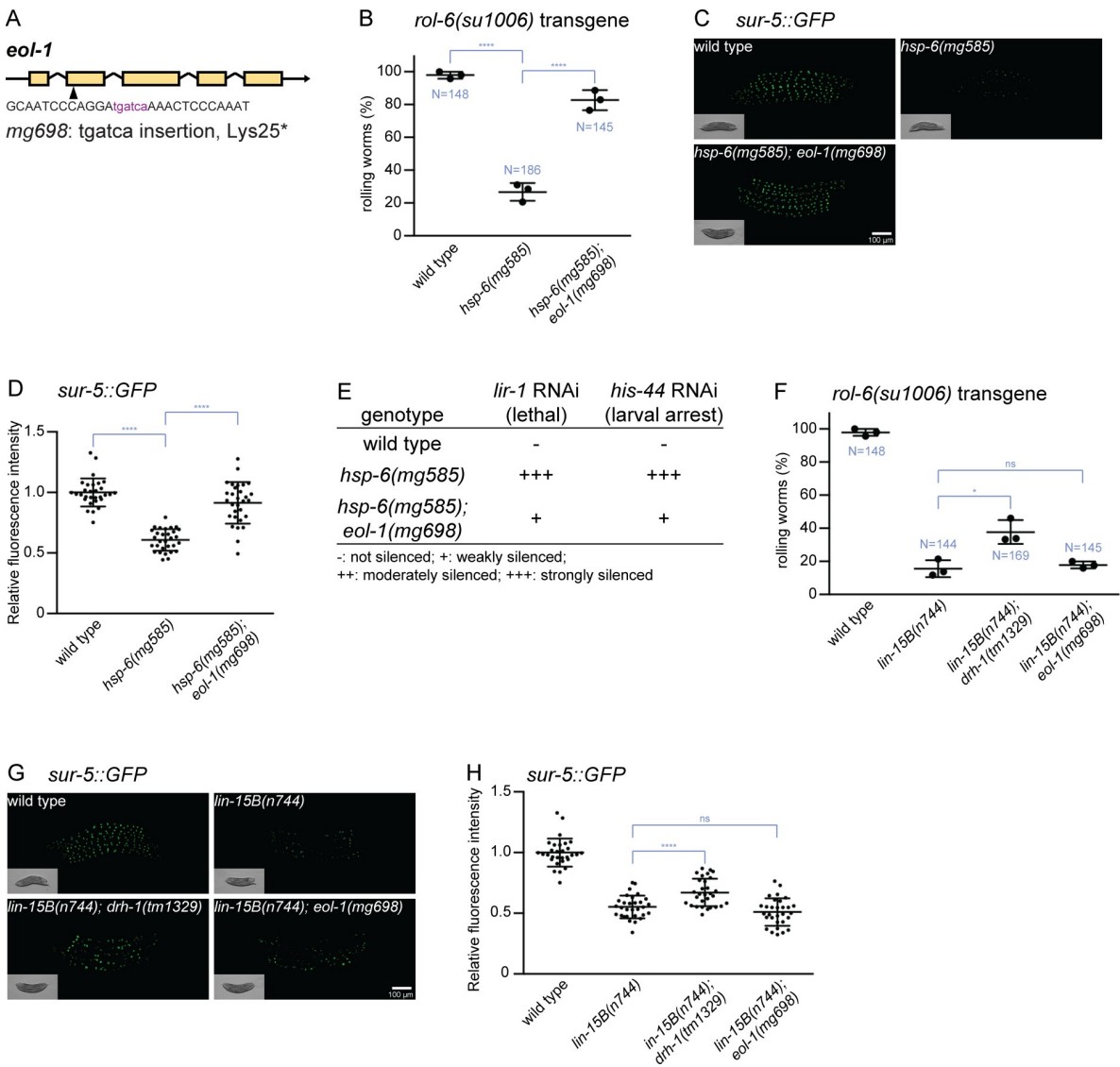

**Fig 3. EOL-1 gene activity is required for Eri.** (A) The *eol-1(mg698)* mutant allele generated by CRISPR-Cas9. (B) Transgene silencing test with the *rol-6(su1006)* multicopy transgene. The expression of the *rol-6* collagen mutation from the transgene is silenced by the Eri in the *hsp-6(mg585)* mutant, and this transgene silencing depends on *eol-1* gene activity. Results of 3 replicate experiments are shown. *N*: total number of animals tested; **** denotes *p* < 0.0001. (C and D) Transgene silencing test with *sur-5::GFP* transgene. This transgene was ubiquitously expressed in all somatic cells. The expression of the *sur-5::GFP* from the transgene is silenced by Eri in the *hsp-6(mg585)* mutant, and this transgene silencing requires *eol-1*. Animals were imaged in (C), and the fluorescence was quantified in (D). **** denotes *p* < 0.0001. (E) Eri response to *lir-1* RNAi or *his-44* RNAi in *hsp-6(mg585)* mutant requires *eol-1*. RNAi of *lir-1* or *his-44* causes lethality/arrest on *hsp-6(mg585)* mutant but not wild type. The Eri is suppressed by the *eol-1(mg698)* mutation. (F) Silencing of *rol-6(su1006)* transgene caused by synMuvB Eri mutations does not depend on *eol-1*. The *rol-6(su1006)* transgene is silenced by the synMuvB *lin-15 (n744)* mutation, and this transgene silencing was slightly suppressed by *drh-1(tm1329)*, but not by *eol-1(mg698)*. Results of 3 replicate experiments are shown. *N*: total number of animals tested; * denotes *p* < 0.05; *ns* denotes *p* > 0.05. (G and H) Silencing of *sur-5::GFP* transgene caused by synMuvB Eri mutations does not require *eol-1*. The *sur-5::GFP* transgene is silenced by the synMuvB *lin-15(n744)* mutation, and this transgene silencing was slightly suppressed by *drh-1(tm1329)*, but not *eol-1(mg698)*. Animals were imaged in (G) and the fluorescence was quantified in (H). **** denotes *p* < 0.0001; *ns* denotes *p* > 0.05. The underlying numerical data can be found in S1 data. Eri, enhanced RNAi; ns, not significant; RNAi, RNA interference.

## Formation of EOL-1 foci requires biogenesis of mitochondrial RNAs

In order to understand the function of DRH-1 and EOL-1 in silencing of transgenes, we monitored their subcellular localization by fusing mScarlet at the N-terminus of DRH-1 and

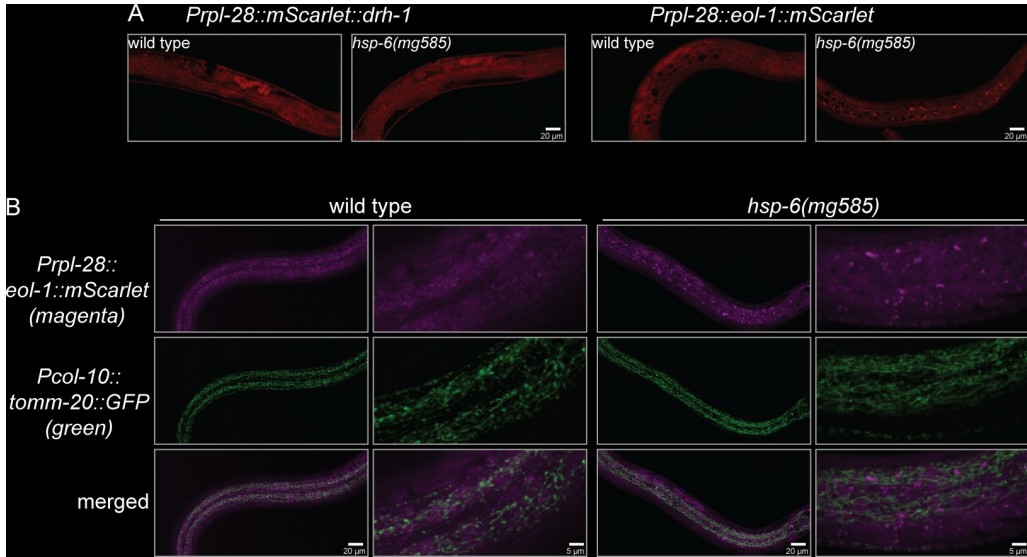

**Fig 4. EOL-1 protein forms cytoplasmic foci if mitochondria are dysfunctional.** (A) EOL-1 protein, but not DRH-1 protein form puncta in the *hsp-6(mg585)* mitochondrial mutant but not in wild type. Mitochondrial dysfunction caused by *hsp-6(mg585)* mutation triggered the formation of EOL-1::mScarlet foci, while mScarlet::DRH-1 remained diffusely localized. (B) EOL-1 foci are not associated with the mitochondria. In the *hsp-6(mg585)* mutant, the EOL-1::mScarlet foci do not colocalize with mitochondria that were visualized by the mitochondrial outer membrane protein TOMM-20:: GFP.

C-terminus of EOL-1, respectively, under the control of *rpl-28* promoter in a miniMOS vector [53]. The miniMOS-generated single copy transgene is able to avoid multicopy transgene silencing by the RNAi pathway, and the ribosome promoter *Prpl-28* is a constitutive promoter with universal expression in all tissues. In wild-type animals, mScarlet::DRH-1 and EOL-1:: mScarlet were localized diffusely in the cytosol without any notable pattern (Fig 4A). In the *hsp-6(mg585)* mitochondrial mutant, strikingly, EOL-1::mScarlet formed massive puncta in many cell types (Fig 4A). mScarlet::DRH-1 remained diffusely localized in *hsp-6(mg585)* (Fig 4A), just as no change was observed in DRH-1::GFP subcellular distribution upon Orsay virus infection [16]. The formation of EOL-1::mScarlet foci in the *hsp-6* mitochondrial mutant, and the specificity of EOL-1 in the suppression of *hsp-6(mg585)* induced silencing, and the mammalian finding that MDA5 recognizes mitochondrial dsRNAs generated a hypothesis that the target of EOL-1 decapping are RNAs derived from the mitochondrial genome. First, we examined whether the cytoplasmic EOL-1 foci localized at the mitochondria surface. Because loss-of-function *eol-1(mg698)* suppresses the mitochondrial defect-induced *rol-6(su1006)* transgene silencing of in the hypodermis, the hypodermal-specific *col-10* promoter-driven TOMM-20:: GFP miniMOS single copy transgene was used to monitor mitochondria in hypodermal cells. However, EOL-1::mScarlet foci were not specifically associated with the mitochondria (Fig 4B). Therefore, EOL-1 accumulates as foci in response to mitochondrial dysfunction, but these foci as not mitochondrially associated and may function in the cytosol.

We explored whether *C. elegans* mitochondria release RNA into the cytosol, as has been observed in the MDA5 mitochondrial surveillance pathway in human and *D. melanogaster* [25,26]. The human mtDNA transcribes 37 genes, including 13 mRNAs encoding subunits of electron transport chain (ETC), 2 ribosomal RNAs (rRNA), and 22 transfer RNAs (tRNA), residing on both the heavy and light strands of the mitochondrial DNA [54]. Transcription on both strands generated 2 genome size polycistronic transcripts that are processed into

A

| Gene | Protein | Fold change in *hsp-6(mg585)* | Gene description |
|------|---------|-------------------------------|------------------|
| *ndfl-4* | ND4L | 4.9 | NADH:ubiquinone oxidoreductase |
| *nduo-6* | ND6 | 4.4 | NADH:ubiquinone oxidoreductase |
| *nduo-3* | ND3 | 3.8 | NADH:ubiquinone oxidoreductase |
| *nduo-5* | ND5 | 3.4 | NADH:ubiquinone oxidoreductase |
| *ctb-1* | CYTB | 3.2 | Cytochrome b |
| *nduo-4* | ND4 | 3.1 | NADH:ubiquinone oxidoreductase |
| *atp-6* | ATP6 | 2.5 | ATP synthase |
| *ctc-3* | COX3 | 2.4 | Cytochrome c oxidase |
| *ctc-1* | COX1 | 2.2 | Cytochrome c oxidase |

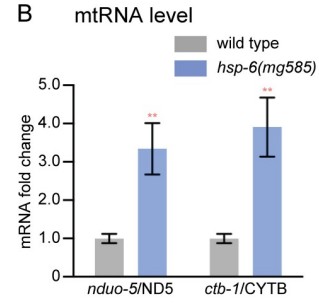

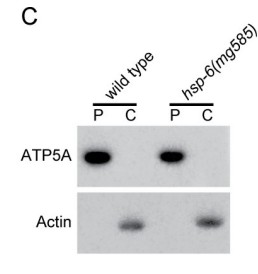

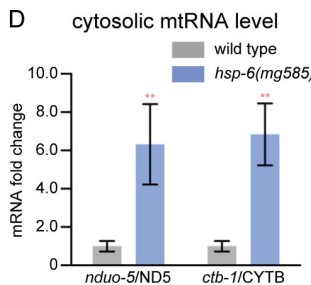

**Fig 5. Release of mitochondrial RNA into the cytosol.** (A) Electron transport chain genes encoded in the mitochondrial genome are induced by the *hsp-6 (mg585)* mitochondrial chaperone gene mutation. Of the 12 genes in the mitochondrial genome that encode subunits of electron transport chain, 9 genes were up-regulated in *hsp-6(mg585)* mutation. (B) The mRNA level of *nduo-5*/ND5 or *ctb-1*/CYTB was induced by *hsp-6(mg585)* mitochondrial mutation. Both *nduo-5*/ND5 (NADH:ubiquinone oxidoreductase) and *ctb-1*/CYTB (cytochrome b) are transcribed from the mitochondrial genome, and their expression level was evaluated by RT-qPCR assays. $^{**}p < 0.01$. (C and D) The mRNA level of *nduo-5*/ND5 or *ctb-1*/CYTB in the cytosol was increased in *hsp-6(mg585)* mitochondrial mutation. The pellet and cytosolic fractions were separated by centrifugation and probed with ATP5A and actin antibodies, respectively, to assess purification in (C). The mRNA level of *nduo-5*/ND5 or *ctb-1*/CYTB in the mitochondria-free cytosolic fraction was evaluated by RT-qPCR in (D). The underlying numerical data can be found in S1 data. The original blot images can be found in S1 raw images. *C*, cytosolic fraction; *P*, pellet fraction; RT-qPCR, quantitative reverse transcription PCR.

individual genes. During this processing, the complementary noncoding RNAs in mammals are degraded by the mitochondrial degradosome formed by RNA helicase SUV3 and polynucleotide phosphorylase PNPase [55]. Loss of SUV3 or PNPase stabilizes the noncoding RNAs to form dsRNAs with the genes derived from the opposite strand [25]. The *C. elegans* mitochondrial genome encodes 36 genes with one less ETC subunit [56]. However, these 36 genes are encoded exclusively on the heavy strand of *C. elegans* mtDNA, and no transcription of the light strand has been detected [57], which disfavors the mitochondrial dsRNA model.

mRNA-seq analysis of *hsp-6(mg585)* revealed that 9 of 12 mitochondrial mRNA encoding ETC subunits were up-regulated (Fig 5A). The increased level of mitochondrial mRNA was further verified by RT-qPCR analysis of *nduo-5*/ND5 and *ctb-1*/CYTB (Fig 5B). In order to examine if the increased abundance of mitochondrial mRNAs within the mitochondrial matrix caused their release into the cytosol, the cytosolic fraction (actin antibody was used as a control protein for this fractionation) was separated by centrifugation from the pellet fraction containing mitochondria (ATP5A antibody used as a control for this fractionation) (Fig 5C). RT-qPCR analyses showed that the mRNA levels of *nduo-5*/ND5 and *ctb-1*/CYTB were substantially increased in the cytosol of *hsp-6(mg585)* compared with wild type (Fig 5D). Analysis of siRNA abundance in *C. elegans* supports the surveillance of mitochondrial mRNAs by the RNAi pathway. Mitochondrially translated mRNAs from the mitochondrial genome generate some of the most abundant siRNAs (S2 Table). For example, the mitochondrial genes *ctc-1*, *nduo-2*, and *nduo-5*, all mitochondrial genome-encoded electron transport chain genes, generate siRNA levels at thousands of reads per million, rivaling those generated by transposable

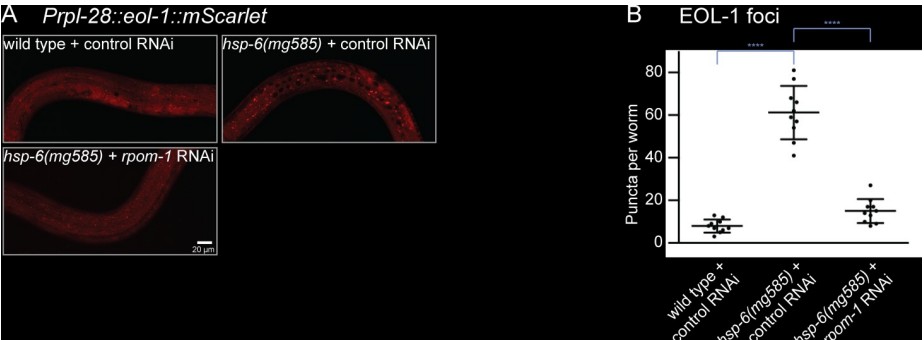

**Fig 6. Formation of EOL-1 foci requires the biogenesis of mitochondrial RNA.** (A and B) Formation of EOL-1 cytoplasmic foci requires the transcription of mitochondrial RNA from mitochondrial genome by RPOM-1 RNA polymerase. The EOL-1::mScarlet foci formed in the *hsp-6(mg585)* mutant were inhibited by RNAi of mitochondrial RNA polymerase *rpom-1*/ POLRMT. Animals were imaged in (A), and the number of foci was quantified in (B). **** denotes $p < 0.0001$. The underlying numerical data can be found in S1 data.

elements in *C. elegans* (the *retr-1* transposable element at rank 24, *ctc-1* at rank 29, *nduo-2* is at rank 104, and *nduo-5* at rank 119).

The strategy of single strand transcription from the mitochondrial genome is not just parochial to *C. elegans*; it is a strategy used by many members of the genus *Caenorhabditis* [58]. If the unique strand transcriptional architecture of the *Caenorhabditis* mitochondrial genome is linked with production of dsRNA from the mitochondria, and if the release of mitochondrial RNA into the cytosol in mitochondrial mutants is the trigger for the formation of EOL-1 foci, we would expect a suppression of EOL-1 focus formation by shutting down mitochondrial RNA biogenesis. The human gene POLRMT encodes a mitochondrial DNA-directed RNA polymerase that catalyzes the transcription of mitochondrial DNA [54]. RNAi of *rpom-1*, the *C. elegans* orthologue of human POLRMT, potently reduced the number of EOL-1 foci in *hsp-6(mg585)* mutant (Fig 6A and 6B). Therefore, the formation EOL-1 foci required the biogenesis of mitochondrial RNA. Mitochondrial RNA polymerase is central to all gene expression in the mitochondrion, including the 12 to 13 protein coding genes, the tRNA genes, and the rRNA genes. So a defect in *rpom-1* is expected to be pleiotropic and thus difficult to ascribe to production of dsRNA from mitochondrial transcripts. But paradoxically, with 36 client genes in the mitochondrion, *rprom-1* has a rather circumscribed sphere of genes it controls, so its function upstream of EOL-1 foci could very well be via dsRNAs produced from mitochondria but released at even higher levels in mitochondrial mutant strains.

## Enhanced RNAi mediates life span extension

While the majority of mitochondrial components are essential for eukaryotes and null alleles of most mitochondrial genes are lethal, *C. elegans* reduction-of-function mitochondrial mutants, including *hsp-6(mg585)* (Fig 7A–7D and S2A–S2D and S2G Fig) increase longevity, in some cases dramatically [28,59,60]. But other mitochondrial mutants, such as *mev-1(kn1)* and *gas-1(fc21)*, shorten the life span (S2E–S2G Fig). We explored whether the Eri output of the *hsp-6(mg585)* mitochondrial mutant contributes to its longevity. *drh-1(tm1329)* or *eol-1(mg698)* single mutants did not shift the survival curve compared with wild type (Fig 7B–7D), showing that these mutants are not short-lived or sickly in some way. But *drh-1(tm1329)* or *eol-1(mg698)* mutations strongly suppressed the life span extension of *hsp-6(mg585)*, showing that *drh-1* or *eol-1* gene activities are required for the extension of life span by *hsp-6(mg585)* (Fig 7C and 7D). Because *drh-1* and *eol-1* mediate the enhanced RNAi, an antiviral response,

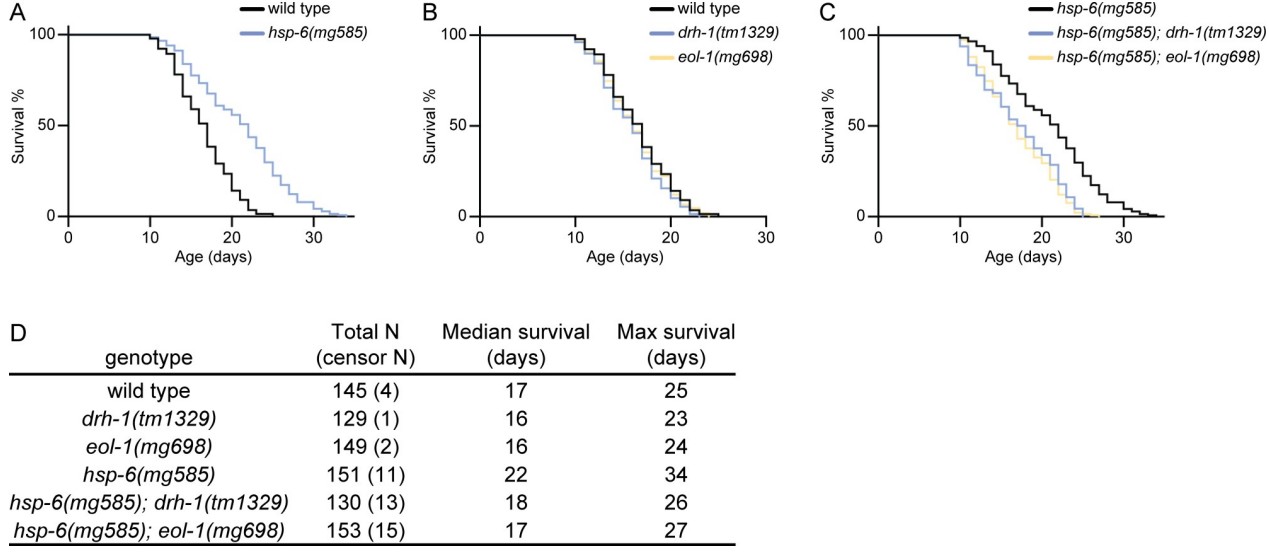

**Fig 7. Decoupling of the Eri response to mitochondrial dysfunction also abrogates the life span enhancement of mitochondrial mutations, but not wild-type life span.** (A) The *hsp-6(mg585)* mitochondrial mutation extends life span. (B) *drh-1(tm1329)* or *eol-1(mg698)* single mutants do not extend or shorten wild life span. (C) *drh-1(tm1329)* or *eol-1(mg698)* suppresses the extension of life span caused by the *hsp-6(mg585)* mutation. (D) Median and maximal survival and number of animals surveyed and censored due to losses during the month-long assay. The underlying numerical data can be found in S1 data.

of mitochondrial mutants, and because these outputs also mediate in the increased longevity of mitochondrial mutants, but not the normal longevity of wild type, these data support the model that enhanced antiviral defense as a key antiaging output from mitochondrial mutations.

## Discussion

### The central role of mitochondria in antiviral immune response

We found that mutations in 4 nuclearly encoded mitochondrial genes *hsp-6*, *nuo-6*, *clk-1*, and *isp-1* strongly enhance RNAi. These mitochondrial mutations enhanced response to dsRNAs introduced by feeding (Fig 1A) and caused the silencing of multicopy transgenes (Fig 1B–1E), a capacity that is strongly associated with enhanced surveillance of foreign genes [36]. NUO-6, GAS-1, and NDUF-7 are components of complex I, MEV-1 is from complex II, ISP-1 is a component of complex III, and CLK-1 produces ubiquinone that transports electrons between complexes I or II and complex III. Three other mitochondrial point mutants, *nduf-7(et19)*, *mev-1(kn1)*, and *gas-1(fc21)* did not enhance RNAi (S1A Fig). Like most mitochondrial mutations, *nduf-7(et19)* lengthens life span (S2D Fig); however, *mev-1(kn1)* or *gas-1(fc21)* shorten life span (S2E and S2F Fig). Mitochondrial components have been identified in previous genome wide screens for Eri and RNAi defective gene inactivations: *coq-1*, a coenzyme Q biosynthetic gene; *aco-2*, aconitase; *idha-1*, mitochondrial isocitrate dehydrogenase; *asb-1* and *asg-1*, subunits of the mitochondrial F0 ATP synthase; *ucr-2.3*, ubiquinol cytochrome c reductase; *tufm-1*, mitochondrial translation elongation factor; *tin-44* mitochondrial inner membrane translocase; and *phb-2* prohibitin, a chaperone for electron transport chain components, constitute a large fraction of the 27 gene inactivations that desilenced transposons in the germline; like viruses, transposons are silenced by RNAi [31,61]. *sdha-2*, which encodes a subunit of complex II, emerged from an Eri screen [36]. There is precedent from other animals for mitochondrial engagement of small RNA pathways, most especially for the piRNAs of Drosophila, which depend on the zucchini mitochondrial phosphatase [62].

Consistent with mitochondrial disruption causing induction of the RNAi pathway, many of the gene inductions in the *hsp-6(mg585)* mitochondrial mutant [22] overlap with those up-regulated by Orsay virus infection (Fig 2A and S1 Table) [43], many of which are the *pals-1* to *pals-40* genes. Mutations in *pal-22* cause an Eri response, suggesting that induction of the *pals* genes may mediate enhanced siRNA-based viral immunity [23,24].

In a viral infection, mitochondria release RNA into the cytosol, as has been observed in the MDA5 mitochondrial surveillance pathway in human and *D. melanogaster* [25,26]. The *C. elegans* mitochondrial genome encodes 36 genes [56] transcribed from one strand [57]. Like mammalian MDA5, we showed that the mRNA levels of *nduo-5*/ND5 and *ctb-1*/CYTB were substantially increased in the cytosol of *hsp-6(mg585)* compared with wild type (Fig 5D). Consistent with the model that dsRNA produced from mitochondrially transcribed and translated mRNAs engage the RNAi machinery in the *C. elegans* cytoplasm, the mitochondrial genes *ctc-1*, *nduo-2*, and *nduo-5*, mitochondrial electron transport chain genes, generate some of the most abundant levels of siRNAs in the genome, similar in abundance to those generated by transposable elements (S2 Table).

Mitochondrial dysfunction may be a key pathogenic feature of viral infection, and systems have evolved to trigger antiviral defense if mitochondria dysfunction is detected. The mitochondria have also been implicated in the Drosophila and mammalian antiviral piRNA pathway: In Drosophila, retroviruses are recognized by the dedicated piRNA pathway that generates piRNAs from integrated flamenco as well as other retroviral elements [63] to target newly encountered viruses that are related to these integrated viruses. A variety of mitochondrial outer membrane proteins couple piRNAs to target loci [64,65]. Similarly, the Sting interferon pathway also uses the MAVS and RIG-I interaction with mitochondria to signal antiviral responses.

Human mitochondrial mutations also trigger antiviral immunity [25,27]. The human RIG-I RNA helicase recognizes viral dsRNA or ssRNA 5′-triphosphate [1]. We showed that mutations in the RIG-1 orthologue *drh-1(tm1329)* suppressed the Eri induced by *hsp-6 (mg585)* (Fig 1). The *eol-1* RNA decapping gene was also strongly induced in virally infected and mitochondrial mutant *C. elegans* in a DRH-1-dependent manner (S1 Table and Fig 2F). The Eri and transgene silencing phenotypes of *hsp-6(mg585)* was suppressed by a *eol-1(mg698)* null mutation (Fig 3E). The localization of the EOL-1 protein was regulated by mitochondrial function: In wild type, EOL-1::mScarlet was localized diffusely in the cytosol (Fig 4A), whereas in the *hsp-6(mg585)* mitochondrial mutant, EOL-1::mScarlet formed massive puncta in many cell types (Fig 4A).

The EOL-1/DXO exonuclease may remove the 5′ cap of mitochondrial RNA, such as a 5′-NAD modification [66], to facilitate the recognition by DRH-1. The 5′ ends of the majority of mRNAs are chemically modified or capped to protect from RNA nucleases [67]. Prokaryotic and eukaryotic cells have different strategies to modify their mRNA with di- or triphosphate in bacteria and m$^7$G cap in eukaryotes. Thus, the eukaryotic 5′ cap is a conspicuous sign to identify self, rather than the non-self-uncapped RNA from virus or bacteria.

The RNAi pathway drives gene silencing of mRNAs that have signatures of being foreign. For example, many of the Eri mutants mediate the silencing of integrated viruses in the *C. elegans* genome that are recently acquired [21]. These viral genes are recognized as foreign by their small number of introns, for example, and by poor splice sites [68]. The antiviral defense of *C. elegans* is more closely related to that of fungi and plants than most animals, for example, Drosophila and vertebrates. Nematodes, plants, and most fungi use the common pathway genes Dicer, specialized Argonaute/PIWI proteins, and most distinctively, RdRPs to produce primary siRNAs and secondary siRNAs against invading viruses. Because these pathways are common across many (but not all) eukaryotes, it is likely that the common ancestor of animals,

plants, fungi, and some protists carried the Dicer, Argonaute, and RdRP genes that mediate RNAi defense.

The loss of RdRP genes in most animal species may be associated with the evolution of the Sting and NF-κB interferon RNA virus defense pathway of most insects and vertebrates, a pathway that is not present in nematodes or plants or fungi [69]. In addition to the interferon pathway, the piRNA pathway has specialized in insects and vertebrates to mediate antiviral defense [70]. The piRNA pathway is not present in fungi or plants and is not as central to viral defense in *C. elegans*, perhaps because it has continued to depend on its RdRP [11].

Viral defense in *C. elegans* is strongly induced in the many mutants that cause enhanced RNAi, *eri-1* to *eri-10*, the many synMuvB mutations, and the mitochondrial mutants we characterized here. These mutations may induce antiviral defense because viral infections may compromise these genetic pathways or because these mutations genetically trigger an antiviral state that is normally under physiological regulation. A surveillance system to monitor the states of these pathways may be an early detection system for viruses to induce expression and activities of Dicer, Argonaute proteins, and RdRPs in siRNA production, targeting, and amplification [71,72]. Regardless of whether antiviral defense is mediated by RdRPs in nematodes or by MAVS and interferon signaling in vertebrates and most insects, the mitochondria are woven into the pathways. For Flock House virus and other nodaviruses, the mitochondrion is a center of viral replication; mutations that compromise the mitochondrion in yeast, for example, cause defects in antiviral responses [73–75].

A decrement in mitochondrial function is one of the most potent mechanisms to increase longevity in a variety of species [28]. *drh-1(tm1329)* or *eol-1(mg698)* mutations strongly suppressed the life span extension of *hsp-6(mg585)*, showing that *drh-1* or *eol-1* gene activities are required for the extension of life span by *hsp-6(mg585)* (Fig 7C and 7D). Because *drh-1* and *eol-1* mediate the enhanced RNAi, an antiviral response, of mitochondrial mutants, and because these outputs also mediate in the increased longevity of mitochondrial mutants, but not the normal longevity of wild type, these data support the model that enhanced antiviral defense as a key antiaging output from mitochondrial mutations.

It is possible that the dramatic increase in frailty in old age could reflect viral vulnerability. In fact, one of the most dramatic hallmarks of the recent Coronavirus Disease 2019 (COVID-19) epidemic is that the virus is far more lethal to the elderly. Conversely, a decrement in antiviral capacity may accelerate aging: In mammals, a reduction in Dicer expression in heart, adipose, and brain is a hallmark of aging [76,77], and a loss-of-function mutant of DCR-1/Dicer shortens *C. elegans* life span [78].

## Materials and methods

### *C. elegans* maintenance and strains used in this study

*C. elegans* strains were cultured at 20˚C. Strains used in this study are listed in S3 Table.

### Generation of transgenic animals

For *Peol-1*::GFP, the plasmid harboring [*Peol-1*::GFP::*eol-1* 3′ UTR + *cbr-unc-119(+)*] was injected into *unc-119(ed3)*. For single-copy transgenes *Prpl-28*::mScarlet::*drh-1*, *Prpl-28*::*eol-1*::mScarlet, and *Pcol-10*::*tomm-20*::GFP, the plasmid was injected into *unc-119(ed3)* following the miniMOS protocol [53]. For CRISPR of *eol-1(mg698)*, we chose *dpy-10(cn64)* as the co-CRISPR marker [52] and pJW1285 (Addgene) to express both of guide-RNA (gRNA) and Cas9 enzyme [79].

## Microscopy

The fluorescent signals of *sur-5::GFP*, *Peol-1::GFP*, and *Phsp-6::GFP* transgenic animals were photographed by Zeiss AX10 Zoom.V16 microscope (Carl Zeiss AG, Oberkochen, Germany). The subcellular pattern of *Prpl-28::mScarlet::drh-1*, *Prpl-28::eol-1::mScarlet*, and *Pcol-10::tomm-20::GFP* were carried out on the Leica TCS SP8 confocal microscope (Leica Camera AG, Wetzlar, Germany). Photographs were analyzed by Fiji-ImageJ (https://fiji.sc/; https://imagej.net/Welcome).

## Sample collection and RNA isolation

For estimation of the mRNA level of *eol-1*, *nduo-5*, and *ctb-1*, around 200 L4 larvae were hand-picked from mixed population and frozen by liquid nitrogen. The total RNA was isolated by TRIzol extraction (Thermo Fisher, Waltham, Massachusetts, United States of America, #15596026). For analyses of cytosolic mRNA level of *nduo-5* and *ctb-1*, approximately 2,000 worms were synchronized by bleach preparation of eggs, hatching progeny eggs to L1 larvae, grown until L4 stage, and frozen by liquid nitrogen. Worm lysates were generated by TissueLyser with steel beads (Qiagen, Venlo, the Netherlands, #69989), and resuspended in 500 μl of the lysis buffer containing 0.8-M sucrose, 10-mM Tris-HCl, 1-mM EDTA, 1× protease inhibitor cocktail (Roche, Basel, Switzerland, 11873580001), and 1-U/μl murine RNase inhibitor (New England BioLabs, Ipswich, Massachusetts, USA, #M0314L). The lysate was centrifuged at 2,500 × g for 10 min at 4°C to remove the pellet debris. And the supernatant was centrifuged at 20,000 × g for 10 min at 4°C. Collect the pellet for western blot, and the supernatant was centrifuged at 20,000 × g for 10 min at 4°C. Take 100 μl of the supernatant for western blot and 300 μl for RNA isolation with TRIzol.

## RT-qPCR

The cDNA was generated by ProtoScript II First Strand cDNA Synthesis Kit (New England BioLabs, #E6560L). qPCR was performed toward *eol-1*, *nduo-5*, and *ctb-1* with *act-1* as control by iQ SYBR Green Supermix (BIO-RAD, Hercules, California, USA, #1708880).

## Western blot

The pellet or supernatant from the previous centrifugation were mix NuPAGE LDS sample buffer (Thermo Fisher #NP0007) and heated at 70°C for 10 min. Samples were loaded onto NuPAGE 4% to 12% Bis-Tris Protein Gels (Thermo Fisher #NP0323BOX) and run with NuPAGE MES SDS running buffer (Thermo Fisher #NP0002). After semi-dry transfer, PVDF membrane (Millipore, Burlington, Massachusetts, USA, #IPVH00010) was blocked with 5% nonfat milk, and probed with anti-ATP5A (Abcam, Cambridge, United Kingdom, ab14748) or anti-actin (Abcam #ab3280) primary antibodies. The membrane was developed with Super-Signal West Femto Maximum Sensitivity Substrate (Thermo Fisher #34096) and visualized by Amersham Hyperfilm (GE Healthcare, Chicago, Illinois, USA, #28906845).

## Life span analysis

Animals were synchronized by egg laying and grown until the L4 stage as day 0. Adults were separated from their progeny by manual transfer to new plates. Survival was examined on a daily basis, and the survival curve was generated by GraphPad Prism (Graphpad Software, San Diego, California, USA).

## Supporting information

**S1 Fig.** (A) Enhanced RNAi response to *lir-1* RNAi or *his-44* RNAi in the *eri-6* control enhanced RNAi mutant or any of the mitochondrial mutants (*nduf-7*, *mev-1*, or *gas-1*), causes lethality/arrest on the mitochondrial mutants but not wild type. (B) Protein alignment of DRH-1 NTD in nematode species. *Ac*: *Ancylostoma ceylanicum*; *Ce*: *Caenorhabditis elegans*; *Cl*: *Caenorhabditis latens*; *Cr*: *Caenorhabditis remanei*; *Nb*: *Nippostrongylus brasiliensis*; NTD, N-terminal domain; RNAi, RNA interference.
(TIF)

**S2 Fig.** (A–D) Mitochondrial mutants (*nuo-6*, *clk-1*, *isp-1*, or *nduf-7*) extend life span. (E and F) Mitochondrial mutants (*mev-1* or *gas-1*) shorten life span. (G) Median and maximal survival and number of animals surveyed and censored due to losses during the month-long assay. The underlying numerical data can be found in S1 data.
(TIF)

**S1 Table. Genes up-regulated in *hsp-6(mg585)* mutant and Orsay virus infection.**
(XLSX)

**S2 Table. siRNA abundances after small RNA isolation from wild-type adult animals in rank order of abundance.**
(XLSX)

**S3 Table. *C. elegans* strains used in this study.**
(XLSX)

**S1 Data. All of the underlying numerical data in this study.**
(XLSX)

**S1 Raw images. Original blot images.**
(PDF)

## Acknowledgments

We thank Caenorhabditis Genetics Center and National BioRescource Project (Tokyo, Japan) for providing strains.

## Author Contributions

**Investigation:** Kai Mao, Peter Breen.

**Supervision:** Gary Ruvkun.

**Writing – original draft:** Kai Mao, Gary Ruvkun.

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
