## [Editor Report · Decision Letter 0]

26 Jun 2020

Dear Gary, 

Thank you for submitting your manuscript entitled "Induction of RNA interference by C. elegans mitochondrial dysfunction via the DRH-1/RIG-I homologue RNA helicase and the EOL-1/RNA decapping enzyme" for consideration as a Research Article by PLOS Biology.

Your manuscript has now been evaluated by the PLOS Biology editorial staff as well as by an academic editor with relevant expertise and I am writing to let you know that we would like to send your submission out for external peer review.

Please re-submit your manuscript within two working days, i.e. by Jun 30 2020 11:59PM.

Kind regards,

Ines

--

Ines Alvarez-Garcia, PhD

Senior Editor

PLOS Biology

Carlyle House, Carlyle Road

Cambridge, CB4 3DN

+44 1223–442810

---

## [Decision Letter · Decision Letter 1]

2 Aug 2020

Dear Gary,

Thank you very much for submitting your manuscript "Induction of RNA interference by C. elegans mitochondrial dysfunction via the DRH-1/RIG-I homologue RNA helicase and the EOL-1/RNA decapping enzyme" for consideration as a Research Article at PLOS Biology. Your manuscript has been evaluated by the PLOS Biology editors, an Academic Editor with relevant expertise, and by two independent reviewers.

As you will see, both reviewers find the conclusions of the manuscript novel and significant for the field, but they also raise a few points that should be addressed mainly to strengthen the results. Reviewer 1 thinks it would be useful to track the mitochondrial RNA using smFISH to see if it’s released to the cytosol. Reviewer 2 suggests two additional experiments mainly to confirm the correlations uncovered in the study.

In light of the reviews (attached below), we will not be able to accept the current version of the manuscript, but we would welcome re-submission of a revised version that takes into account the reviewers' comments. We cannot make any decision about publication until we have seen the revised manuscript and your response to the reviewers' comments. Your revised manuscript is also likely to be sent for further evaluation by the reviewers.

We expect to receive your revised manuscript within 2 months. 

**IMPORTANT - SUBMITTING YOUR REVISION**

*Re-submission Checklist*

*Published Peer Review*

*PLOS Data Policy*

*Blot and Gel Data Policy*

Sincerely,

Ines

--

Ines Alvarez-Garcia, PhD,

Senior Editor,

ialvarez-garcia@plos.org,

PLOS Biology

Reviewers’ comments

Rev. 1: Oded Rechavi – please note that this reviewer has waived anonymity

This manuscript was a pleasure to read and I think it would be of interest to many readers from different fields, and especially to researchers studying RNAi. Briefly (I won't summarize all the discoveries that they made), they show that worms sense the state of their mitochondria, and that upon disruption of mitochondrial functions they mount an antiviral RNAi response. They follow the mechanism and find that how it involves eol-1, drh-1, and potentially secretion of dsRNA to the cytosol. I like the paper as is but here are some small suggestions:

Introduction:

- The introduction might benefit from more references. For example, the entire part about RdRP is unreferenced. This could be important because, for instance, the paragraph that contains these sentences otherwise sounds very speculative: "RNA viruses of course depend on such RdRp genes for their replication so the defense pathway may have evolved from the viral weapon, or vice versa. The fact that RdRp genes mediate RNAi functions across eukaryotic phylogeny suggests that these pathways have been lost for example in most of the vertebrate lineages that only have the first stage of RNAi, the Argonautes and Dicer, rather than RdRps evolving or being acquired by horizontal transfer from viruses independently in so many eukaryotic lineage" - There's literature about this, for example from Eugene Koonin's lab. I am not an RdRP expert but as far as I know viral RdRPs and the RdRPs of animals evolved independently (I was under the impression that unlike viral RdRPs, the RdRPs of animals evolved from DNA dependent RNA polymerases).

Results:

- Is it possible to track the mitochondrial RNA using smFISH to see if it's released to the cytosol? I think this part is less convincing, but the authors don't write this part carefully, so it's ok (They only say that the process "might" involved secretion of dsRNA). To show directly an increase (using smFISH) in cytosolic RNA would help substantiate the claim.

- Of course one could dig further to study how drh-1 and eol-1 mediate the lifespan extension of the mitochondrial mutants, but I think this is beyond the scope of the paper. This part, as is, feels a little out of tune with the rest of the paper, because this is a major discovery (the link to aging) which is not explored (again because it's out of scope which is ok). Perhaps this could be improved by moving this to the discussion or expanding on the theory. If I understand correctly the proposed mechanism is a tradeoff of increased longevity at the price of vulnerability to viruses at old age?

- The figures in the PDF that I downloaded from the PLoS Bio site are in very low resolution (probably something about the upload to the site) that makes it hard to assess certain observations (most notably in the microscopy pictures, e.g. aggregation of eol-1).

discussion:

the discussion could be more organised, perhaps it would help to break it into subsections.

I think that it's worth expanding on the link between this paper and the previous papers from the Ruvkon lab about danger signals in C.elegans (e.g. Melo and Ruvkon, Cell 2012). This paper is referenced but i think it could be discussed further, because conceptually it's highly related: worms sense their internal state (in this case mitochondrial functions) and RNAi is part of the surveillance mechanism.

Small typo: the abbreviation RdRP is written in different ways across the paper (RdRp, Rdrp).

Rev. 2: Roy Parker – note that this reviewer has also waived anonymity

In this article, Mao and colleagues describe a process whereby mitochondrial dysfunction [hsp-6(mg585) allele] in nematodes enhances RNAi and triggers antiviral gene induction. The authors show that this process is dependent on DRH1 (a homolog of the mammalian antiviral RIG-I/MDA-5 proteins) and EOL1 (a paralog of the mammalian antiviral Dxo protein). The authors show that hsp-6(mg585) results in increased cytoplasmic levels of mitochondrial RNAs, which is known to trigger the MDA-5/RIG-I-mediated interferon response in mammalian cells. Moreover, the synthesis of mtRNA is required for the formation of ELO-1-containing cytoplasmic foci in response to hsp-6(mg585)-mediated mitochondrial dysfunction. Lastly, the authors show that the hsp-6(mg585) mitochondrial mutant increases longevity, and this increase is dependent on both DRH-1 (RIG-I/MDA-5) and EOL-1 (Dxo). The authors thus propose that the DRH-1/ELO-1 enhanced antiviral defense pathway provides an anti-aging mechanism in response to mitochondrial mutations.

The finding that hsp-6(mg585) is stimulating a RIG-I/MDA-5-like innate antiviral response in C. elegans is an interesting finding since C. elegans lack the MAVS-interferon antiviral response. Thus, this finding provides insight into previous, orthologous and/or alternative functions of mammalian interferon response proteins mediated by RIG-I and MDA-5. These finding also show that mitochondrial dysfunction is a conserved mechanism between mammal and C. elegans by which antiviral proteins sense viral infection. Lastly, the finding that EOL-1 is required for this function and forms cytoplasmic foci is interesting, although the significance of these foci remains unclear.

This review is from Roy Parker and I would be willing to clarify these comments for the authors if needed.

Major Comments:

1) As it stands, there is a correlation that some mutations increase mitochondrial dysfunctions, increase RNAi, and increase mitochondrial RNA in the cytosol. The significance of this correlation would be strengthened if they demonstrated that the mitochondrial mutants that do not increase RNAi, do not increase the presence of mitochondrial RNA in cytosol.

2) As it stands, there is a correlation that some mutations increase mitochondrial dysfunction, RNAi, and lifespan. To show this correlation is pertinent to the increased lifespan, the authors should show that mitochondrial dysfunction that does not increase RNAi does not increase lifespan.

3) Since there is a substantial literature on how mitochondrial dysfunction increases lifespan (e.g. Zhang et al., 2018, Cell), the work should be put in the context of earlier studies in this area.

Additional comments:

4) The title, abstract, and discussion state DRH-1 activity as homologous to RIG-I, but in the results section DRH-1 is primarily stated as an MDA-5 homolog. The authors should provide clarity on this discrepancy since RIG-I and MDA-5 have different RNA substrate preferences in mammalian cells. Based on homology, is DRH-1 expected to have substrate preferences more closely aligned to MDA-5 or RIG-I substrate preferences?

5) References to Figure 5 preceded references to Figure 4 in the text. Moreover, Figure 4 references break up references to Figure 5 sub figures. This is confusing. The authors should consider re-arranging Figures/text to make this section easier to navigate for readers.

6) I found the difficult to read, primarily due to too much background information throughout the article, and complicated paragraph structure. This could be improved.

---

## [Editor Report · Decision Letter 2]

21 Oct 2020

Dear Gary,

Thank you for submitting your revised Research Article entitled "Induction of RNA interference by C. elegans mitochondrial dysfunction via the DRH-1/RIG-I homologue RNA helicase and the EOL-1/RNA decapping enzyme" for publication in PLOS Biology. I have now discussed the revision with the Academic Editor and the rest of the team. 

We're delighted to let you know that we're now editorially satisfied with your manuscript. However, we would like you to consider changing the title to "Mitochondrial dysfunction induces RNA interference in C. elegans through a pathway homologous to the mammalian RIG-I antiviral response."

Before we can formally accept your paper and consider it "in press", we also need to ensure that your article conforms to our guidelines. A member of our team will be in touch shortly with a set of requests. As we can't proceed until these requirements are met, your swift response will help prevent delays to publication. Please also make sure to address the data and other policy-related requests noted at the end of this email.

- a cover letter that should detail your responses to any editorial requests, if applicable

*Copyediting*

*Published Peer Review History*

*Early Version*

Best wishes,

Ines

--

Ines Alvarez-Garcia, PhD,

Senior Editor,

ialvarez-garcia@plos.org,

PLOS Biology

Fig. 1B, E; Fig. 2E, F, H; Fig. B, D, F, H; Fig. 5B, D; Fig. 6B; Fig. 7A-C and Fig. S2A-F

Please also provide a figure legend for Suppl. Table 3.

For manuscripts submitted on or after 1st July 2019, we require the original, uncropped and minimally adjusted images supporting all blot and gel results reported in an article's figures or Supporting Information files. We will require these files before a manuscript can be accepted so please prepare and upload them now. Please carefully read our guidelines for how to prepare and upload this data: https://journals.plos.org/plosbiology/s/figures#loc-blot-and-gel-reporting-requirements.

---

## [Editor Report · Decision Letter 3]

9 Nov 2020

Dear Dr Ruvkun,

On behalf of my colleagues and the Academic Editor, René F. Ketting, I am pleased to inform you that we will be delighted to publish your Research Article in PLOS Biology. 

PRODUCTION PROCESS

Before publication you will see the copyedited word document (within 5 business days) and a PDF proof shortly after that. The copyeditor will be in touch shortly before sending you the copyedited Word document. We will make some revisions at copyediting stage to conform to our general style, and for clarification. When you receive this version you should check and revise it very carefully, including figures, tables, references, and supporting information, because corrections at the next stage (proofs) will be strictly limited to (1) errors in author names or affiliations, (2) errors of scientific fact that would cause misunderstandings to readers, and (3) printer's (introduced) errors. Please return the copyedited file within 2 business days in order to ensure timely delivery of the PDF proof. 

If you are likely to be away when either this document or the proof is sent, please ensure we have contact information of a second person, as we will need you to respond quickly at each point. Given the disruptions resulting from the ongoing COVID-19 pandemic, there may be delays in the production process. We apologise in advance for any inconvenience caused and will do our best to minimize impact as far as possible.

EARLY VERSION

PRESS 

Kind regards,

Alice Musson

Publishing Editor, 

PLOS Biology

on behalf of

Ines Alvarez-Garcia,

Senior Editor

PLOS Biology